# Anti-Bourgeois Media in the Japanese Proletarian Literary Movement

**Takashi Wada**

Faculty of Education, Mie University, Mie 514-8507, Japan; wadataka@edu.mie-u.ac.jp

**Abstract:** The Marxist and socialist ideas that spread throughout the world following the Russian Revolution of 1917 were also influential in bringing about changes in art and culture. Proletarian literature, which flourished in Japan in the 1920s and 1930s, was one such example. However, due to Japan's particular historical circumstances, Japanese proletarian literature was in an ambivalent position between revolutionary literature by left-wing intellectuals and proletarian literature by and for the proletarian class in the pure sense. This article examines the chaos and friction implied by the term "proletarian" from three perspectives: the relationship between proletarian media and bourgeois media, the media distribution system, and the boundary between writers and readers. Through this examination, it clarifies that the approaches of Japanese proletarian media, while imitating bourgeois media to some extent, were unique in their potential to transform the boundary between writers and readers.

**Keywords:** Marxism; leftist; censorship; modernism; Walter Benjamin

## 1. Introduction

The proletarian literature that flourished in Japan in the 1920s and 1930s was somewhat different from that of other countries. In Japan's most famous encyclopedia of modern literature, Odagiri Hideo explains that "proletarian literature" "refers not so much to proletarian literature as to the totality of the socialist and communist revolutionary literature of the late *Taishō* and early *Shōwa* periods. Therefore, the concepts of 'labor literature' and 'workers' literature' do not completely overlap with the concept of 'proletarian literature' as such, but are related to each other, with labor literature being of a pre-proletarian period and workers' literature being an important part of proletarian literature." (Odagiri 1977, p. 451). Thus, Japanese proletarian literature, due to Japan's particular historical circumstances, was in an ambivalent position between revolutionary literature by left-wing intellectuals and proletarian literature by and for the proletarian class in the pure sense.[1] To begin with, the word "proletariat" in the dictionary itself has a broad scope, as it is explained as "That class of the community which is dependent on daily labour for subsistence, and has no reserve or capital; the indigent wage-earners; sometimes extended to include all wafe-earners; working men, the labouring classes" (Simpson and Weiner 1989, p. 606). The polysemy of proletarian literature is therefore not limited to Japan. For example, Barbara Foley, who studied American proletarian literature, argues that proletarian literature was variously defined by Marxist commentators in terms of the class background of the author, the class background of the readership, the nature of the subject matter, or the perspective adopted on that subject matter (Foley 1993, pp. 86–128). In Japan, however, although there was much discussion about creative methods, there was little debate about what defines proletarian literature. In other words, whereas in the USA the very meaning of the term proletarian literature was a disputed issue, in Japan the term itself was used in a stable manner.[2]

Regarding the ambivalent yet firm use of proletarian literature in Japan, Nakano Shigeharu, who was party to the proletarian literary movement, asserts that this was mainly due

to the fact that the word "revolution" was banned in pre-war Japan.[3] The pre-war *Dainippon Teikoku Kenpō* 大日本帝国憲法 (Constitution of the Empire of Japan, promulgated in 1890) clearly stipulated freedom of expression, but only in principle: as Article 29 states, "Japanese subjects shall have freedom of speech, writing, publication, assembly, and association within the limits of the law", with censorship imposed according to "law". Laws governing censorship of literary works included the *Shinbunshi-hō* 新聞紙法 (Newspaper Act, 1893) and *Shuppan-hō* 出版法 (Publishing Act, 1909). Censorship under these laws was initially carried out before publication; by the 1920s, however, it was being performed after publication. Therefore, editors of newspapers and magazines used *fuseji* 伏字 (starring out dangerous terms) so that they would not be banned or otherwise penalized by ex post facto censorship.[4] These circumstances led to the use of the term "proletarian" as a substitute for "revolution".

Many of the intellectuals who led the proletarian literary movement were highly educated and influenced by Western culture, which Japan had used as a model for modernization, or by the Japanese bourgeois culture that had emerged from its reception.[5] The relationship between these intellectuals and the laboring farmer masses, who were to be the subjects of the revolution but were not intellectually educated, contained potential for a variety of cultural chaos and friction over literary literacy. For example, the *geijutsu taishūka ronsō* 芸術大衆化論争 (controversy on the popularization of art) starting in 1928 (a frequent topic of discussion within the proletarian literary movement on how to disseminate proletarian culture to workers and peasants) often led to divided opinion between critics, who were elite intellectuals, and writers, who were from the working class.[6]

The chaos and friction surrounding the conflict between bourgeois culture and proletarian culture in the proletarian literary movement may also have been expressed in the media through which proletarian works were published and in the relationship between authors and readers mediated thereby. In this paper, therefore, I examine the relationship between the proletarian literary movement and bourgeois culture from the perspective of media theory.

## 2. The *Henshū Sayoku* Proletarian Literature Boom

The proletarian literary movement that flourished in Japan from the late Taishō era to the early Shōwa era (1920s–1930s) still has some weight in the history of Japanese literature. If the readership of proletarian literature had been limited to those involved in the revolutionary movement, this literature would not be as well-known as it is today. In fact, proletarian literature had its fair share of readers and was commercially viable at the time.

The heyday of proletarian literature from a commercial point of view was between 1928 and 1930. The "Overview of the Publishing World" section of the *Shuppan Nenkan* 出版年鑑 (Publishing Yearbook), often cited as a reference for the success of left-wing publications during this period, notes that in 1929, "Marxist books and proletarian novels continued to flourish as in the previous year", (Tokyo-dō 1930, p. 11) and in 1930, "quite a few were blindly involved in left-leaning books out of a rambling, vague business strategy that anything with a leftward tendency would sell without regard for either nation or society in the face of greed." (Tokyo-dō 1931, p. 13) In 1931, the situation changed: "On the ideological front as well, the fashionable left leanings have somewhat declined this year, while the serious ones have gone into hiding and disappeared." (Tokyo-dō 1932, p. 15)[7] The majority of left-wing publications in Japan were books related to proletarian literature, as the 1929 edition describes them as "Marxist books (i.e., history books, etc.)" and "proletarian novels (i.e., literary books)." In light of these descriptions and the rush to publish the *Sōsho* 叢書 (series of books) discussed below, 1930 can be considered the peak of the proletarian literature boom.

This boom in proletarian literature was brought about by the bourgeois media. Proletarian literary works were sometimes published in major newspapers and especially in the two major general magazines, *Kaizō* 改造 and *Chūō Kōron* 中央公論, which actively carried the works of proletarian writers. Around 1930, there were two factions in the Japanese pro-

letarian literary movement: the *Bungei Sensen* 文芸戦線 (Literary Front) faction, which supported the *Rōnō-tō* 労農党 (Labor-farmer Party), the legal proletarian party, and the *Senki* 戦旗 (Battle-flag) faction, which supported the Communist Party, which was illegal until the end of WWII.[8] Although *Kaizō* and *Chūō Kōron* published proletarian works regardless of faction, these two general magazines were rivals competing for sales: the former carried relatively more works from the *Bungei Sensen*, while the latter actively published works from *Senki*, considered more radical in its link to the illegal party.[9]

One of the reasons why these general magazines actively published proletarian literature was the *henshū sayoku* 編集左翼 (editorial left) approach, daring to make their magazines radical in order to sell. Former *Chūō Kōron* editor-in-chief Amemiya Yōzō recalled an episode in which, after the January 1929 issue was banned for publishing proletarian writer Kuroshima Denji's "*Hyōga (Glacier)*" about the Siberian exodus, the president, Shimanaka Yūsaku, estimated that "this would lead to sales" despite the economic blow it would cause to the company. According to Amemiya, publishing proletarian literature "was a nerve-wracking experience for the editors, but at the time, the editorial line was so close to the ban that it was the way to get the magazine onto the commercial market, both ideologically and in terms of public morals. Even with liberalism at heart, the *henshū sayoku* took shape there of itself." (Amemiya 1988, p. 536).

However, although these bourgeois media actively made use of proletarian writers, the mainstream of their fiction writing columns was dominated by so-called bourgeois writers such as Shimazaki Tōson and Tanizaki Junichirō.[10] Amemiya cited not only the sharp writing and the background of the labor–management conflict, but also the fact that the manuscript fees for proletarian writers were "about 3 yen compared to the average 8 yen for so-called bourgeois writers", (Amemiya 1988, p. 537) as reasons for publishing proletarian literary works. Herein lay the paradox that the writers of proletarian literature, who sought to expose the injustice of the exploitation of workers and peasants by capitalists and landlords, were themselves exploited by the profit-oriented bourgeois media as producers of works used for their cost performance.

Proletarian literature was not only published in magazines but was also consumed as books. In Japan, for example, some media published complete works and selected editions, labeling proletarian literature and the *Shin Kankaku-ha* 新感覚派 (New Sensibility School) as *shinkō* 新興 (modern) or *sentan* 尖端 (cutting-edge), seeing them as a co-occurrence of modernism literature. *Shin Kankaku-ha* was a literary group with formalist tendencies that emerged in the mid-1920s and was opposed to proletarian literature, as if Japanese proletarian literature could be compared to the Russian *На посту* (Na postu, Marxist) and *Shin Kankaku-ha* to *Леф* (Lef, Avant Garde). To cite a specific example, in *Gendai Nihon Bungaku Zenshū No. 50: Shinkō Bungaku Zenshū* 現代日本文学全集第 *50* 巻：新興文学全集 (Complete Modern Japanese Literature, 50th Arc: Complete Modern Literature, Tokyo: Kaizo-sha, 1929), short stories by Maedako Hiroichiro, Hayama Yoshiki, and Kataoka Teppei of proletarian literature were included along with those by Kishida Kunio and Yokomitsu Riichi of the *Shin Kankaku-ha*.[11] In addition, the 28-volume *Shin'ei Bungaku Sōsho* 新鋭文学叢書 (New Literature Series, Tokyo: Kaizo-sha, 1930) and the 20-volume *Shin Geijutsu-ron System* 新芸術論システム (New Art Theory System, Tokyo: Tenjin-sha, 1930–1931) published the works and theories of proletarian literature and *Shin Kankaku-ha* as part of the same series; the 14-volume *Sekai Daitokai Sentan Jazu Bungaku* 世界大都会尖端ジャズ文学 (World Metropolitan Cutting-edge Jazz Literature, Tokyo: Shun'yō-dō, 1930–1931) considered both proletarian literature and the *Shin Kankaku-ha* to come under its purview, albeit in different volumes.

Thus, proletarian literature rode the wave of the proletarian literary boom created by the bourgeois media industry, increasing its circulation and readership. However, the bourgeois media was more concerned with profit than ideology and, moreover, tended to blur the distinction between proletarian literature and other literatures.

### 3. Imitating and Differentiating from the Bourgeois Media

The fact that many works by proletarian writers were published and disseminated in general magazines such as *Kaizō* and *Chūō Kōron* was not necessarily welcomed by proletarian literary organizations: most of the readers of these magazines were in the intellectual class, and there were not many workers and peasants, the movement's ideal readers. Therefore, the challenge for the proletarian literary movement was how to be read not only by intellectuals but also by the laboring farmer masses.[12]

Of course, this is not to say that workers and farmers did not read general magazines. Rather, their ratio increased with the rise in educational levels from the *Taishō* to the *Shōwa* era. According to Nagamine Shigetoshi, with the emergence of workers who read, a "pyramid of readers" was formed, with intellectual readers at the top and worker readers at the bottom. A certain number of workers had a desire for self-education, of which general magazines such as *Kaizō* and *Bungei Shunjū* 文芸春秋 took advantage to adopt a low-priced, mass-market approach, leading to the emergence of readers among workers who moved from the bottom to the top of the pyramid. However, despite their popularization, most of the contents of the general magazines were geared toward students and salarymen, and there was an insurmountable qualitative gap between intellectual readers and worker readers (Nagamine 2001, pp. 161–201).

Therefore, it was necessary for proletarian literature to form its own anti-capitalist media (let us call it "proletarian media"). As for magazines, however, since *Tanemakuhito* 種蒔く人 (The Sower) (Feb. 1921–Aug. 1923), the dawn of Japanese proletarian literature, a wide variety of proletarian magazines had been published in the form of coterie magazines and journals. *Bungei Sensen* and *Senki*, mentioned earlier in the rivalry between the general magazines *Kaizō* and *Chūō Kōron*, were also the names of magazines, as explained in Note 6. However, although these magazines were distinct in terms of their content, which included editorials and literary works from the standpoint of the communist movement and the workers and peasants, their layout and other aspects were modeled after the format of the bourgeois magazines.

As proletarian literature gained momentum in the late 1920s, the proletarian media began to publish books as well as magazines. These were the publishers *Bungei Sensen-shuppanbu* and *Senki-sha*. *Senki-sha*, in particular, developed remarkable publishing activities that swept the bourgeois market, publishing Japanese representative works of proletarian literature such as Kobayashi Takiji's *Kanikōsen* in 1929 and Tokunaga Sunao's *Taiyō no Nai Machi* in 1929 as inexpensive books in a series called *Teihon Nihon Puroretaria Sakka Sōsho* 定本日本プロレタリア作家叢書 (Japanese Proletarian Writers Series).[13]

That said, the publishing strategy of the *Teihon Nihon Puroretaria Sakka Sōsho* was similar to that of the *en-pon* 円本 (one-yen books), a successful series developed by the bourgeois media at the time.[14] The *en-pon* began as a large-scale sale of 25 volumes of the *Gendai Nihon Bungaku Zenshū* for one yen each in 1926 by the publisher *Kaizō-sha*, which advocated the liberation of art (literature) from the privileged class to the people. Thereafter, other publishers followed suit. Moreover, some believe that it was leftist publications that played a role in this *en-pon* boom.[15] In other words, the *Teihon Nihon Puroretaria Sakka Sōsho* imitated the bourgeois media's method of gaining a mass readership.[16]

However, the proletarian media had one characteristic that the bourgeois media did not: their method of distribution and sales. In Japan, prewar and present alike, the general distribution and sales of books and magazines involves a wholesaler who acts as an intermediary to distribute the publications to retailers nationwide. However, because of the direct or indirect contributions of the proletarian media to the socialist revolution in terms of content and funds, they were constantly monitored by the authorities and often banned under a censorship system that conformed to the Japanese laws of the time (the Newspaper and Publishing Acts). Therefore, in order to ensure that the magazine could reach readers even in the event of a ban, *Senki-sha* organized branch offices and reading circles, establishing a distribution network through which the magazine could be sold directly to readers.

Tsuboi Shigeji called this method of distribution to each retail outlet nationwide via a distributor a street-corner or bourgeois distribution network, referring to the method of distribution directly to readers through branch offices or reading circles as a direct distribution network, which enabled distribution even in the face of a ban. (Tsuboi 1931, p. 194) That is to say, the latter method was similar to modern-day subscriptions. Also, through this method, *Senki* grew from a circulation of 7000 as of its first issue in May 1928 to a circulation of 22,000 by its 24th issue in March 1930. (Tsuboi 1931, pp. 195–96).

One of the factors that contributed to the success of *Senki*'s direct distribution network may have been the peculiar reading systems of the workers of the time. Unable to afford the magazines and books they wanted due to low wages, they often lent and borrowed magazines and books purchased by someone else (or jointly bought) to circulate.[17] For example, Matsumoto Seicho, who became one of Japan's leading mystery writers after the war, borrowed *Senki* from his artisan friends through this distribution network when he was young (Matsumoto 1977, pp. 261–62). Within the direct distribution network established by *Senki-sha*, organizing reading circles, in particular, was a very effective strategy because it took advantage of the characteristics of workers who "shared" books.

These strategies were not, however, original to *Senki-sha*. The branch office system was already in place in the *MusanshaShinbun* 無産者新聞 (Proletarian Newspaper), which was launched in 1925 and served as the legal organ of the then-illegal Japanese Communist Party (Nimura 1978, pp. 9–12). As for reading circles, the popular magazine *King* (founded in 1925), with an ideology opposite to that of the proletarian media, had already organized these associations. Nagamine, who analyzed the readership of *King*, points out that *King* was accepted not only individually by the atomized masses but also communally and collectively in relation to others: *King*'s reading groups were characterized by the fact that they were organized "vertically" from top to bottom, led by leaders of youth groups, the military, schools, companies, and other local communities and intermediary organizations, in addition to the spontaneous "horizontal" organization among readers that was common in other magazines (Nagamine 1997, pp. 222–39). The proletarian media *Senki*'s reading circles were developed in opposition to these "local communities and intermediary organizations" by replacing them with factory and rural "unions"[18].

## 4. Cultivating Writers in the Proletarian Media

Thus, the proletarian literary movement was clearly conscious of different sales methods from those of the bourgeois media. In April 1930, the Japan Proletarian Writers' League (JPWL) issued a declaration entitled "*Our Attitude toward Bourgeois Publications Must be Like These.*" The JPWL was formed in 1929 as the main group under the aforementioned *Senki* faction and was strongly influenced by Russian revolutionary literary theory. As such, this declaration imitated the principle of writing by party members for publications outside the party, which the prerevolutionary Russian Social Democratic Labor Party had adopted in November 1907, which attached the following conditions to the use of bourgeois publications:

1. Participation in bourgeois publishing should be limited to a secondary use, such as earning a living or financing activities, and participants should not be under the illusion that these works can agitate workers or peasants.
2. Participation in the editing of bourgeois publications shall be only "when we have full control" or "for technical work only (as in proofreading)".
3. In view of our responsibility to the proletariat for any work we do, we must never use aliases or anonymity when submitting manuscripts for publication in bourgeois publications.
4. For *tankōbon* 単行本 (single-volume hardcover books), the limited editorial intervention and the small number of publishing houses available to us require that we examine the nature of the publishing house before using it (Nihon Puroretaria Sakka Dōmei Chūō Iinkai 1930a, pp. 178–79).

This declaration stipulated that writers belonging to the Writers' Union should focus on enriching the proletarian media rather than the bourgeois media, and that the bourgeois media should be used only under various conditions.

It is noteworthy that this provision was submitted at about the same time as the "*Resolution on the Popularization of Art*" (June 1930) by the same JPWL.[19] As mentioned earlier, the JPWL had been arguing over the popularization of art since 1928, and this resolution brought an official decision. The resolution, as relevant to this paper, called for proletarian writers to enter into the real life of the masses as a search for a form of proletarian art, and also identified the germ of this new form in the *Rōnō Tsūshin* 労農通信 (Labor-farmer News), in which the working peasant masses throughout Japan contributed to one another.[20] In other words, the massification that the proletarian literary movement was to aim for by focusing on the proletarian media called first of all for writers to descend from the top down to experience the real lives of the masses, as well as for workers and peasants themselves to sublimate their own creative methods from the bottom up. Indeed, the JPWL published a new medium, the twice-monthly *Bungaku Shinbun* 文学新聞 (Literary Newspaper, 10 Oct. 1931–5 Oct. 1933), which was based on the *Rōnō Tsūshin*.

This possibility of the boundary between writer and reader being transformed as writers approach the masses using newspapers as a medium of practice and the masses themselves becoming producers of literature is consistent with Walter Benjamin's proposal in his work "*The Author as* Producer" (1934). Benjamin, drawing inspiration from the "author as 'operator'" as defined and embodied by the Soviet playwright and journalist Sergei Tretyakov, explains:

> For as writing gains in breadth what it loses in depth, the conventional distinction between author and public, which is upheld by the bourgeois press, begins in the Soviet press to disappear. For there the reader is at all times ready to become a writer—that is, a describer, or even a prescriber. As an expert—not perhaps in a discipline but perhaps in a post that he holds—he gains access to authorship. Work itself has its turn to speak. (Benjamin [1934] 1999, p. 771)

However, the attempts of the *Bungaku Shinbun* did not move toward reforming the concept of literary forms and genres, as Benjamin had suggested. Rather, while bearing the potential to dismantle traditional forms and genres, it can be seen as having reinforced them.

One cause of it was the political bias that subordinated the literary movement to the organizational activities of the Communist Party.[21] The JPWL was oriented toward separating the *Rōnō Tsūshin* from literature, seeing it solely as part of its political activities. Akita Ujaku, a leading proletarian writer in Japan who had traveled to the Soviet Union, repeatedly emphasized that "the movement of the labor correspondents is never about creating literature" (Akita 1930, p. 101) and that "the work of the labor correspondents must first be an accurate report of the life of the workers and peasants rather than the production of literature" (Akita 1931, p. 71). Nakano Shigeharu, another proletarian writer, echoed and summarized Akita's assertion as follows: "The correspondents' movement has its own purpose and is not for the sake of literature" (Nakano 1931, p. 36). That is to say, in the JPWL, the *Rōnō Tsūshin* system was used as a tool to expand the membership of the organization.

The potential for the creation of new literary forms was destroyed not only by this division between *Rōnō Tsūshin* and literature, but also by the reinforcement of traditional literary concepts. This is symbolized by the prize system established in *Bungaku Shinbun*.[22] The *Bungaku Shinbun* established a prize system specifically for fiction writing, in addition to the contributions submitted by workers and peasants throughout the country. On the first page of the 6th issue (5 Jan. 1932), which carried the winning entries of the "New Year Issue Special Prize", the biggest project of all, the following were highlighted: "Literature by the workers and peasants has been born" and "Now Japanese proletarian literature has moved to factories and farming villages!!"

Let us ignore for the moment the fact that the system of awarding prizes for fiction works was reproduced in the bourgeois media; even so, there are two problems with awarding prizes in the proletarian media.[23] First, the prize clearly separated "fiction writing" from the other contribution columns, making the selection and publication of the submitted work itself an objective for readers. This tendency was already evident in the contributions that appeared in the *Bungaku Shinbun*, many of which were submitted under real names rather than anonymously, even though they referred to politically dangerous content such as labor disputes. Since the selection committee for the special prize in the 6th issue of *Bungaku Shinbun* was composed of prominent proletarian writers such as Kobayashi Takiji and Tokunaga Sunao, it is thought that many literary youth from the working and peasant classes wanted their recognition. Ultimately, it was possible that proletarian literature, which was supposed to sublimate literature and politics, became a tool for the self-realization of the desires of literary youth. Of course, there were also writers who turned to practical social activities as a result of these desires. Kobayashi, who was one of them, submitted many of his writings to bourgeois media such as *ShōsetsuKurabu* 小説倶楽部 (Novel Club) and *Shinkō Bungaku* 新興文学 (Modern Literature), and when his play *Jo-shūto* 女囚徒 (Female prisoners) was first published in the proletarian media, he expressed his pure pleasure in his diary.[24] However, the prize system was only a device for the birth of writers, not for the creation of new literary forms.

Another problem with the prize system was that the winning entries received a review, preserving the traditional relationship between the writer selecting and the reader being chosen. Moreover, *Bungaku Shinbun* also published "*Shōsetsu no Kakikata* 小説の書き方" (How to Write a Novel, No. 6) and "*Wareware no Bungaku-kōza* われわれの文学講座" (Our Literature Lectures, Nos. 13 and 15), indicating a clear hierarchy between writers and readers of the laboring farmer masses. Furthermore, these "How to Write" and "Lectures" articles belied their names: the contents tended to evoke the superiority of workers and peasants, such as "Workers and peasants can become good writers only by working in the workplace and fighting against the bourgeoisie" ("*Shōsetsu no Kakikata*" in *Bungaku Shinbun*, No.6: 4).

This tendency of successful proletarian writers to instruct the laboring farmer masses in spiritual theory can also be seen in the "*Shōsetsu Saku-hō* 小説作法 (Novel Writing Method)" of Tokunaga Sunao and Kobayashi Takiji, which was published around the same time in the *Sōgō Puroretaria Geijutsu Kōza* 綜合プロレタリア芸術講座 (Comprehensive Proletarian Art Course, 1931). For example, Tokunaga writes, "Writing (expression) can be explained in outline, but the essential part is **Discipline** and **Learning**." (Tokunaga 1931, p. 142). Kobayashi writes, "To tell the truth, there is no such thing as a 'way of creating' a novel." (Kobayashi 1931, p. 99), and then adds, "A proletarian writer must above all make this Marxist point of view his own. To this end, you must **cultivate yourself rigorously**." (Kobayashi 1931, p. 111). In short, the cliché of proletarian fiction writing was that "discipline" or "cultivation" were necessary to become a proletarian writer.

Interestingly, this approach to novel instruction was consistent with *Shōsetsu Saku-hō* (1909) by Tayama Katai, a leading naturalist writer during the *Meiji* and *Taishō* eras. He began this book by stating, like Kobayashi, that "There is no such thing as method in novels", and that "in fact, there is no such thing as method other than patience and discipline" (Tayama 1909, p. 1). As the editor-in-chief of the literary contribution magazine *Bunshō Sekai* 文章世界 (Writing World, 1906–1920), Tayama was a very influential figure among young men of letters who aspired to pursue a literary career. Tayama's emphasis on "discipline" as a novel writing method resonated with the *shūyō-shugi* 修養主義 (cultivationism) of the time. *Shūyō-shugi*, a term in opposition to *kyōyō-shugi* 教養主義 (culturism), meant that young people who had been excluded from academic courses cultivated a morality that was different from knowledge through self-study and self-discipline. In brief, for literary youths, writing a good novel and having it published in a magazine or becoming a writer was a form self-realization as an alternative to the academic course, so Tayama encouraged "discipline" for this purpose.[25]

The readers of the proletarian media were also mostly laborers and peasants who had been excluded from the academic track. For them, too, writing for the *Bungaku Shinbun* and winning the paper's prizes were part of their self-realization. The editors and writers of the proletarian media were able to arouse the self-assurance of the laboring farmer masses that they could write good works by staying in the field of labor; by teaching them novel-writing methods that offered little in the way of techniques, they mass-produced writers who could not enter the bourgeois media but were published in the proletarian media, thereby increasing the number of readers and participants in the social movement. This is similar to the situation in which many of the literary youth who wrote for *Bunshō Sekai* continued to be reproduced as readers without becoming professional writers.

## 5. Conclusions

As discussed above, the Japanese proletarian literary movement, while advocating an anti-bourgeois media, developed in complicity with the bourgeois media and sought to differentiate itself therefrom while imitating their know-how. The anti-capitalism that was the movement's goal was not only political, but also a revolutionary attempt to overthrow the media system and literary genres themselves. On the other hand, *Bungaku Shinbun*, which attempted to dismantle traditional literary forms and genres, cannot be said to have achieved this. By distinguishing between contribution writing and fiction writing through calls for prizes, and by having well-known writers review and instruct readers on writing, it preserved the traditional boundaries between writers and readers. While the magazine aimed at differentiation, it is undeniable that the result was a repetition of the existing system.

However, there was a slight glimpse in the *Bungaku Shinbun* of the possibility of the kind of change that Benjamin proposed: its layout. As mentioned earlier, the *Bungaku Shinbun* received many contributions under real names, but the contributions sent by workers and peasants from all over the country were treated on the same level as proletarian writers who already had readers in commercial magazines: articles by unknown contributors and correspondents often appeared next to those of famous writers. The magazines, and likewise proletarian media, did not do this. In magazines, the names of write-in contributors and correspondents appeared in small font in the table of contents, with their articles clearly differentiated from those of famous writers by being arranged in columns. The *Bungaku Shinbun* editorial policy, which called on the workers and peasants themselves to become writers, and the characteristics of the newspaper medium intertwined with each other to demonstrate the possibility of revolutionizing the boundary between writers and readers in terms of the paper's layout.

It is easy to criticize the lack of literary quality in proletarian literature on the basis of political superiority. In fact, contrary to its "*Bungaku* (=Literature)" name, the *Bungaku Shinbun* eventually began to carry political articles, and it too became biased toward the extreme left.[26] But it should not be overlooked that during the brief period between 1930 and 1931, when the proletarian literary boom reached its peak and then entered its decline, there was a small possibility, albeit unfulfilled, that the established literary forms and genres might be transformed. This paper has attempted to reevaluate the situation from the perspective of media theory.

**Funding:** This research was funded by JSPS KAKENHI Grant Numbers JP19K13053, JP22K00338.

**Conflicts of Interest:** The author declares no conflict of interest.

## Notes

1	For example, Ragon (1986) discusses not revolutionary writers in the broad sense, including intelligentsia, but the activities of pure "proletarian writers" in France, drawing a clear line. In addition, while there was an organization called the Japan Proletarian Writers' League, the name of a similar organization in Germany was the "*Bund proletarisch-revolutionärer Schriftsteller* (Proletarian and Revolutionary Writers' League)", using the word "revolution".

2    As Foley (1993, p. 95) notes, "For in discussions of proletarian authorship it was often not clear [ . . . ] whether the term "proletarian author" denoted a "radical" or "revolutionary" proletarian author or simply one "from the proletariat", the meaning relating to the word "proletariat" was continuously debated in the USA.

3    Nakano (1946, pp. 29–31) offers, among other reasons for the name "proletarian literature", that in prewar Japan the proletarian class had to take over the finishing touches of the bourgeois democratic revolution, making the strategic error of making the leap to a proletarian revolution without going through a bourgeois democratic revolution.

4    A good English-language reference on the censorship system in modern Japanese literature is Rubin (1984).

5    This perception has become common knowledge in Japanese literature studies, with Uranishi (2001, p. 53), a famous scholar of proletarian literature, noting that, "as is well known", the NAPF faction of the largest force in proletarian literature groups (the *Senki* faction, discussed below) "was dominated by intellectuals with a higher education".

6    Wada (2009) discusses the conflict of opinion between intellectual critics and writers from working-class backgrounds.

7    Umeda (1998, pp. 22–23) cites three reasons for the rapid decline of left-wing publications after the 1930s: the more stringent censorship system, the over-competition of left-wing publications, and the growing exclusionism of public opinion since the Manchurian Incident.

8    Both *Bungei Sensen* and *Senki* were the names of proletarian literary magazines. The former (Jun. 1924–Dec. 1930) was originally a larger concern, launched in 1924 when the ties between political parties and writers were still weak, but after several subsequent alliances and ruptures, *Senki* (May 1928–Dec. 1931) was launched in 1928 and overtook its progenitor.

9    According to statistics in Kurihara (2004, pp. 97 and 247), *Kaizō* and *Chūō Kōron*'s fiction sections published "29 PL [proletarian literature], 3 EA [emerging artists], and 68 EW [established writers] in the period from April 1929 to March 1930, and 49 PL, 8 EA, and 55 EW in the period from April 1930 to March 1931". The *Shinkō Geijutsu-ha* (Emerging Artists), who were hostile to proletarian writers, were often published in literary magazines such as *Bungei Shunjū* and *Shinchō*, meaning that proletarian literature had no meaningful lead, but at least in general magazines, its publication rate was dramatically high.

10   Shimazaki Tōson, from a venerable merchant family, was one of leading naturalist writers in Japan and the first president of the Japan PEN Club (1935–1943). Tanizaki Junichirō, from a wealthy Tokyo merchant family, was a leading writer of aesthetic literature in Japan.

11   Kishida (1925) denied that "I had ever called myself part of the *Shin Kankaku-ha*", but it was the label applied to his work by the journalism of the time.

12   Kurihara (2004, pp. 253–54) points out that the romantic sympathy of the masses, mainly the intelligentsia, for the communist movement and organization made Marxism and class struggle one of the fads of the time, with the radical seen as "cool": the majority of the readers of proletarian literature were not "factory workers" or "peasants", but these intellectual fellow travelers.

13   From 1928 to 1929, the *Bungei Sensen-sha* published eight books including *Seryōshitsu nite: A Collection of Short Stories* by Hirabayashi Taiko in 1928, while from 1929 to 1931, *Senki-sha* published 11 volumes of *Teihon Nihon Puroretaria Sakka Sōsho* and three volumes of *Nenkan Nihon Puroretaria Shishū* 年刊日本プロレタリア詩集 (Yearly Japanese Proletarian Poetry Collections), the 1929, 1931, and 1932 editions. In addition, these two companies were not in fact the first to publish proletarian media: they were preceded by *Tanemakuhito*'s *Tanemaki-sha*. However, although this company planned to publish a total of five volumes of the *Hito to Shisō Sōsho* 人と思想叢書 (People and Ideology Series), it ended up publishing only one volume, Kaneko Yobun's *Ikiteiru Mushanokoji Saneatsu* in 1922.

14   Maeda (1973, pp. 240–42) points out that in 1927, before the organization of proletarian literature had settled into the two factions discussed here, the influence of the *en-pon* could already be seen in the way proletarian magazines were published.

15   Sato (2002, pp. 67–68) notes that the leftist publications playing a role in the *en-pon* boom included *Marxism Sōsho* in 28 volumes (*Kōbun-dō*), *Marx-shugi Kōza* in 13 volumes (*Ueno Shobō*), and the world's first *Complete Works of Marx and Engels* in 27 volumes (a joint project among five publishers).

16   Kimura (2022, p. 142) evaluates the historical significance of *Senki-sha*'s series publication, saying that its orientation in literary history differed from "publishing capitalist" literary history. While I generally agree with Kimura's argument, the proletarian media remained aligned with the conventional style in their imitation of the bourgeois media format.

17   Nagamine (2001, pp. 160–70), through an analysis of a reading survey of approximately 14,000 factory workers in Tokyo in 1935, notes that factory workers overwhelmingly borrowed books from "acquaintances" as their most frequently used reading device, and points out that "for the workers, books existed not as something owned individually but rather as something shared in a mutual loan relationship among their peers".

18   Sato (2002, p. 69), noting that magazines for boys and women were published as separate editions of *Senki*, and that a nationwide organization of readers called *Senki Shikyoku* 戦旗支局 (*Senki* Branch-office) was created, states that the publication of *Senki* "can be described as a counter movement that directly challenged the mass public nature of *King*".

19   It is important to note the contradiction here: "As the issue of popularization was being discussed, the cultural organizations put the brakes on their use of the 'bourgeois magazines' most visible to the 'masses' and shifted their activities toward confinement in the booming space of leftist publishing." (Tatemoto 2020, p. 150).

20 The Nihon Puroretaria Sakka Dōmei Chūō Iinkai (1930b, pp. 166–76) argued that the writer "must first and foremost plunge into the real life of the masses and there grasp the practical basis of our own form." In the *Rōnō Tsūshin*, a contribution "written by the struggling proletariat to report the state of their activities to a wide circle of comrades from factories, farming villages, and all other scenes of struggle", there was hope for a new proletarian literary form that would be attuned to the real life of the masses.

21 For a discussion of the process by which the *Bungaku Shinbun*, initially planned as an extension of the popularization of literature, fell into a far-left bias against the backdrop of organizational theory by the Communist Party, see Uranishi (1989).

22 In the *Bungaku Shinbun* (33 issues in all), a "Call for Prizes" can be found in No. 1 (10 Oct. 1931), No. 3 (20 Nov. 1931), No. 4 (5 Dec. 1931), and No. 9 (20 Feb. 1932). In the smaller contests, the winning entries were published in the "Readers' Literature" columns of the 2nd (1 Nov. 1931), 3rd, 8th (5 Feb. 1932), and 13th (25 Apr. 1932) issues, and the prizes were relatively inexpensive, such as a magazine copy or book vouchers. Even in the "New Year Special Prize", which offered the highest prize money, the six winning novelists received only 5 yen each, the winning poet 10 yen, and the three honorable-mention poets 3 yen each. This was at a time when the starting salary for elementary school teachers was around 50 yen per month.

23 As discussed in Kōno (2003, p. 26), the awarding of prizes for novels was also done in Europe and the US, but in Japan it took root on its own as "an event of publishing capital focusing on the selection process itself in the media".

24 Kobayashi ([1927] 1993, p. 131), "Diary of 25 Aug. 1927": "Significantly, my '*Joshūto*' (one-act play) is to appear in the October issue of the *Bungei Sensen*! This is the magazine that is now the center of attention throughout the literary scene." The entry describes the excitement of the event.

25 See Yamamoto (2014), who points out that Tayama's *Shōsetsu Saku-hō* emphasized process rather than result, discussing the resonant relationship between the "patience and discipline" espoused by Tayama and the *shuyō-shugi* of the time.

26 Yamada (1932) introduced protests from reading circles against the conversion of the *Bungaku Shinbun* into a political newspaper.

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
