# Peer review of "Anti-Bourgeois Media in the Japanese Proletarian Literary Movement"

_humanities, doi:10.3390/h11060160_

Round 1

Reviewer 1 Report

This is an exceptionally well-researched and wide-ranging piece. It provides helpful context for the international reader and a broad scope for the benefit of future scholars. It also furnishes interesting discussion of the fluctuations in popularity of Japanese proletarian literature and the material challenges facing working-class writers and their publishers.

The point about the problematic issues facing the proletarian writers’ movement, as articulated from lines 258-262, needs to be unpacked and perhaps restructured into the two paragraphs that follow: make the essential points, link them more securely to the foregoing discussion, and then elaborate on them in a more integrated way in the two passages where you discuss ideological/organisational restrictions. The piece also discusses key elements in terms of what might be termed “cultural capital” and “social capital”, yet it never gets to grips with how this has been theorised in relation to cultural production by thinkers like Jacques Rancière and Pierre Bourdieu—key thinkers on class and literature. Some greater theorisation would be welcome here and elsewhere, including on the distinctions between genres that the author discusses.                                                                                                               

Restructure lines 27-29 for clarity. Lines 262-264 could be more elegantly expressed e.g. “The purpose of our current discussion is…”. On line 320, replace “was” with “were”.

Otherwise, this is a fascinating and hard-working piece and I will look forward to seeing it published.

Reviewer 2 Report

General appreciation: 

This paper deals with the following topic : “Anti-Bourgeois Media in the Japanese Proletarian Literary Movement”. Howeverthe author of this paper does not explain clearly what the originality of this contribution in the field is. Key concepts are not clearly defined. Moreover, the author must indicate Japanese expressions both in Hepburn romanization and in Japanese.

Consequently, this paper needs to be improved before any publication.  

Some points that must be improved:

Key concepts such as “petty-bourgeois intelligentsia” or “proletarian literature” are not defined. What are their meaning in Western scholarship? Do such categories fit to Japanese context? Are these categories mobilized for hermeneutic purposes or are they completely efficient? 

The expression “bourgeois media industry” (line 19) should be explained. What does it include? What does it exclude?

Some expressions are presented as merely synonymous, but the relationship between them must be questioned or justified (for instance, “proletarian literature” and “left-wing publications”).

Some statements need to be supported by an academic source (lines 32-33, 34-36). 

The name of some organizations is indicated without any historical explanation about their creation, their importance, and their action (for instance, lines 206-207).

Since Humanities is not a “Japanese studies journal”, Japanese writers must be presented further in a note (line 93).

Some explanations require a Japanese language ability to be understood (for instance, lines 373-374). We would suggest indicating the Japanese when the Hepburn romanization is indicated. 

To conclude, the author needs to clarify some explanations and be more accurate before any publication.  

Round 2

Reviewer 2 Report

The author followed our suggestions and improved his paper, which is greatly appreciated. 

We would propose some small final improvements listed below:

- When the Japanese writing is indicated, it would be easier towards non-Japanese speakers to indicate first the Hepburn romanization and, just after, the Japanese writing (and not the opposite).

- Please note that the Hepburn romanization of プロレタリア is puroretaria (lines 191, 359, 461, 462, etc), of ジャズ is jazu (line 150), of 倶楽部 is kurabu, of 修養主義 is shūyōshugi. 

- line 20, the hyphen in "total-ity" is not necessary.  

- line 418, check the spelling of "literature"

- Please check both the title and its romanization line 465.

- line 447, some characters seem weird. 

- line 464, I think that there are some unnecessary blanks here. 

- line 540, "Histoire" and not "Historie".

I think it could be useful to allow the author to take 10 more days to check everything. 

Best wishes,
